# Effects Induced by the Temperature and Chemical Environment on the Fluorescence of Water-Soluble Gold Nanoparticles Functionalized with a Perylene-Derivative Dye

**DOI:** 10.3390/ma17051097

**Published:** 2024-02-28

**Authors:** Agnieszka Lindstaedt, Justyna Doroszuk, Aneta Machnikowska, Alicja Dziadosz, Piotr Barski, Vittoria Raffa, Dariusz Witt

**Affiliations:** 1ProChimia Surfaces Sp. z o.o., Zacisze 2, 81-850 Sopot, Polandjusdor3101@gmail.com (J.D.); aneta.machnikowska@gmail.com (A.M.); alicja.dziadosz.ad@gmail.com (A.D.); office@prochimia.com (P.B.); 2Dipartimento di Biologia, Università di Pisa, S.S. 12 Abetone e Brennero, 4 56127 Pisa, Italy; vittoria.raffa@unipi.it; 3Faculty of Chemistry, Gdansk University of Technology, Narutowicza 11/12, 80-233 Gdansk, Poland

**Keywords:** gold nanoparticles, gold nanorods, temperature-dependent fluorescence, perylene-derivative dye

## Abstract

We developed a fluorescent molecular probe based on gold nanoparticles functionalized with *N*,*N*′-bis(2-(1-piperazino)ethyl)-3,4,9,10-perylenetetracarboxylic acid diimide dihydrochloride, and these probes exhibit potential for applications in microscopic thermometry. The intensity of fluorescence was affected by changes in temperature. Chemical environments, such as different buffers with the same pH, also resulted in different fluorescence intensities. Due to the fluorescence intensity changes exhibited by modified gold nanoparticles, these materials are promising candidates for future technologies involving microscopic temperature measurements.

## 1. Introduction

Temperature is among the most important parameters that reflect the energy of a system. By measuring the temperature, we can at least partially characterize some physical or chemical process in the living or still environment. For living cells, temperature changes are indicative of disease pathologies associated with disturbed pathways for energy production and consumption. The methods used to measure temperature at the macroscopic scale are well known and understood. These methods involve analog thermometers based on liquid expansion, thermocouples, infrared probes, and thermal imaging cameras. However, it is challenging and complicated to measure temperature at the microscopic scale, such as the temperature of objects just a few micrometers in size, such as cells. In the last decade, several fluorescent molecular probes have been developed to determine the temperature at the microscopic scale [1]. Through these probes, the temperature inside microreactors, cells, or living organisms can be measured. By determining the temperature with high special and temporal resolution, we can understand micro reactions and rationally develop effective therapies at the cellular scale.

Generally, there are five principal methods used to measure temperature-dependent fluorescence. The first and most common method is based on changes in emission intensity [2,3,4,5,6]. The second method to measure temperature is based on the ratiometric fluorescence of a molecular probe [7,8,9,10,11,12,13,14,15,16]. The third approach involves changes in the lifetime of the probe [17,18]. The fourth method involves measuring the ratio of emission peaks of the two conjugated fluorophores that exhibit different responses to temperature [19,20]. The fifth method follows the emission peak shift vs. temperature [21,22,23,24,25,26,27]. Each method exhibits advantages and disadvantages. Most frequently, photobleaching [28], sensitivity to ionic strength, pH [2,29], and concentration dependence are observed. The observation of changes in emission intensity provides a rapid method for determining the temperature. However, each measurement must be calibrated with a baseline. Moreover, the results are greatly affected by photobleaching and the variation in the concentration of probes in each cell [2,3,5,7]. From that point of view, the ratiometric self-calibrated method is more attractive. However, these thermal probes are often polymers and can be affected by ionic strength or pH [8]. Moreover, their large dimensions [2,9,12] can be responsible for the limited internalization, diffusion, and disturbance of cell functions. The major advantage of using the lifetime, a ratio of emission peaks, or the emission peak shift as the temperature indicator is that the results are independent of thermal probe concentration. For the method based on the lifetime measurement of the probe, photobleaching is not a problem, as the bleached fluorophores are not measured. Although several molecular probes have been obtained to determine the temperature, further studies are needed to develop reliable, widely available, commercial molecular thermometers.

Perylene dyes are an important class of chromophores. Due to the properties of these dyes, several applications with perylene dyes have been developed thus far. These dyes have been used as fluorescence standards [30,31,32], thin film transistors [33,34,35], liquid crystals [36,37,38], light emitting diodes [39], and photovoltaic devices [40,41,42]. The perylene molecule consists of five benzene rings fused together to provide an extended π-conjugated planar structure. A variety of perylene dyes can be readily obtained when perylene tetracarboxylic acid dianhydride is used as the starting material [43,44,45]. Moreover, water-soluble perylene diimide derivatives were produced to provide potential antitumor drugs, fluorescence tags, and elements of self-assembled photoactive films [46,47,48,49,50,51]. In this study, to explore a new reversible micro thermometer, gold nanoparticles (AuNPs) functionalized with *N*,*N*′-bis(2-(1-piperazino)ethyl)-3,4,9,10-perylenetetracarboxylic acid diimide dihydrochloride (PZPER) were synthesized, and their reversible optical properties were investigated.

## 2. Materials and Methods

Sodium citrate (cat# 71402), EDCI (cat# E7750), sodium *N*-hydroxysulfosuccinimide (cat# 56485), AgNO_3_ (cat# 209139), perylene-3,4,9,10-tetracarboxylic dianhydride (PTCDA) (cat# P11255), L-ascorbic acid (cat# A92902) benzylhexadecyldimethylammonium chloride (BDAC) (cat# B4136), hexadecyltrimethylammonium bromide (CTAB) (cat# 52365), 1-(2-aminoethyl)piperazine (cat# A55209), and an Amicon ultra centrifugal filter (50 kDa, MWCO) (cat# UFC9050) were purchased from Sigma Aldrich (St. Louis, MO, USA). HAuCl_4_ × H_2_O (cat# 12325) and HSPEG3000COOH (cat# PEG1099) were purchased from Alfa Aesar (Waltham, MA, USA) and IRIS Biotech (Marktredwitz, Germany), respectively.

### 2.1. Preparation of N,N′-bis(2-(1-Piperazino)ethyl)-3,4,9,10-perylenetetracarboxylic Acid Diimide Dihydrochloride (PZPER)

PZPER was obtained using a previously described method [52]. Perylene-3,4,9,10-tetracarboxylic dianhydride (PTCDA) was treated with excess 1-(2-aminoethyl)piperazine and acidified with 2 M HCl. The reaction product was precipitated as the hydrochloride salt in acetone. The spectra were in agreement with the previously reported data [52].

### 2.2. Preparation of Gold Nanoparticles and Nanorods

Gold nanoparticles (AuNPs, 12 nm) stabilized with citrate were prepared according to a previously described method [53]. Spherical nanoparticles were obtained by reducing gold chloride using sodium citrate in hot water.

Gold nanorods (AuNRs, 8.8 × 39.8 nm) stabilized with BDAC and CTAB were obtained, as described previously [54]. The gold nanorod AuNRs were produced by seed-mediated growth method based on a CTAB-capped seed. The silver content of the growth solution was used to grow NRs to a desired length. The experimental procedure is provided in the Appendix A (Appendix A).

### 2.3. Preparation of Functionalized Gold Nanoparticles

A solution of HSPEG3000COOH (2.5 µmol) in DI water (2 mL 0.2 µm CA filter) was added to 50 µmol of 12 nm Au citrate-stabilized nanoparticles. The mixture was stirred for 1 h at RT. Then, the mixture was transferred to an Amicon filter 50 kDa and centrifuged for 20 min using a bench top centrifuge (2300 rpm). Centrifugation was continued until the minimum retention volume reached ca. 200–300 µL. Mixing by pipetting and washing was repeated 3 times with DI water using a centrifugal filter (Amicon 50 kDa). This step aims to remove excess free HSPEG3000COOH. AuNPs were suspended in DI water (10 mL), and the full characteristics (UV–Vis, DLS, zeta potential) were determined.

To 10 µmol of Au_12_PEG3000COOH, a solution of EDCI (0.38 µmol) in DI water (1 mL) was added, and then the mixture was stirred for 5 min at RT. Next, a solution of *N*-hydroxysulfosuccinimide sodium salt (0.38 µmol) in DI water (1 mL) was added, and the mixture was stirred for 20 min at RT. Then, PZPER hydrochloride (0.19 µmol) was added to the mixture and stirred for 1 h at RT. The mixture was transferred to an Amicon filter 50 kDa, and the samples were centrifuged for 20 min using a bench top centrifuge (2300 rpm). Centrifugation was continued until the minimum retention volume reached ca. 200–300 µL. Mixing by pipetting and washing was repeated 5 times with DI water using a centrifugal filter (Amicon 50 kDa). AuNPs were suspended in DI water (2 mL), and the full characteristics (UV–Vis, DLS, zeta potential) were determined.

### 2.4. Determination of Size and Zeta Potential

A dynamic light scattering (DLS) Zetasizer Ultra (Malvern Panalytical Ltd., Malvern, UK) analyzer was used to determine the hydrodynamic particle size and zeta potential of AuNPs. The detection angle was set at 90°, the temperature was set at 25 °C, and the refractive index was set at 1.33. A helium–neon laser beam and a clear polystyrene cuvette (3 mL, 10 × 10 × 45 mm) were used for sample analysis.

### 2.5. Transmission Electron Microscopy (TEM)

The size and morphology of the prepared AuNPs were determined using a JEM 1400 (JEOL Co., Tokyo, Japan, 2008) with an energy-dispersive full range X-ray microanalysis system (EDS INCA Energy TEM, Oxford Instruments, London, UK), a tomographic holder, and an 11 Megapixel TEM Camera MORADA G2 (EMSIS GmbH, Münster, Germany) (JEOL, Japan; Laboratory of Electron Microscopy, the Nencki Institute of Experimental Biology Polish Academy of Science, Warsaw, Poland). The samples were prepared by placing them on carbon-coated copper and leaving it to air-dry before imaging.

### 2.6. UV–Vis and Fluorescence Measurement

UV–Vis spectroscopy was performed using a Lambda 365 spectrophotometer (PerkinElmer, Waltham, MA, USA). Fluorescence spectra were recorded by a FL 6500 fluorescence spectrometer (PerkinElmer, USA).

## 3. Results and Discussion

The presence of protonated peripheral nitrogen atoms improves the solubility of PZPER in water and polar organic solvents and potentially provides a chromophore that can be highly modulable as a function of pH, temperature, and the dielectric constant of the solution. The optical properties of the dye were first characterized in aqueous solution, and spectral changes at different temperatures were examined (Figure 1).

Although minor changes vs. temperature at 540 nm (UV–Vis) can be observed (Appendix A, Appendix A, Appendix A), for AuNPs functionalized with PZPER, the amount of chromophore will likely be insufficient to observe a reliable response vs. temperature. Moreover, the plasmon resonance of gold nanoparticles will overlap the expected signal from PZPER. The problems with the gold background can be overcome by employing the fluorescence properties of PZPER. In this case, AuNPs cannot disturb the response of fluorescence probes to temperature. We examined the fluorescence of an aqueous solution of PZPER (Figure 1).

### 3.1. Effects Induced by the Temperature and Buffers on the PZPER Fluorescence

As presented in Figure 1, the changes in the intensity of emission at 590 nm can be related to the changes in temperature. However, every fluorescent probe must be tested for variables such as polarity, pH, viscosity, and molecular interactions before application in complex media/systems. We tested the fluorescence of PZPER in water and buffers (10 mM, pH 7.4), PBS, Tris, and MOPS at 25 and 80 °C. The results are summarized in Figure 2a.

Although the largest changes are observed for water, the pH cannot be controlled. The quenching of the signal was the smallest for the MOPS buffer. Further experiments were performed in this buffer at different concentrations. We verified the fluorescence of PZPER at 25 and 80 °C in MOPS (1 mM and 10 mM, pH 7.4). When the concentration of PZPER (50 μM) is relatively low, a concentration of 1 mM of buffer should be sufficient to maintain a stable pH. The spectra are presented in Figure 2b (spectra in water are added for comparison).

A lower (1 mM) concentration of MOPS buffer provided a higher intensity of the PZPER signal. It can be expected that a higher concentration of any salt can quench fluorescence as well. The optimal buffer was MOPS at a 1 mM concentration. We also examined this buffer at different pH values (5.35 and 7.4) (Figure 3a). Spectra in water were added for comparison.

Surprisingly, the signal for PZPER in MOPS buffer (1 mM) was the strongest at 80 °C and was the same as that in water at 25 °C when the buffer pH was 5.35. The comparison of the spectra at 25 °C, pH 7.4 (green) and pH 5.3 (violet), suggests that changes in the fluorescence intensity are related to changes in the pH (Figure 3a). Determining the optimal conditions for the detection of PZPER can be helpful in the case of functionalized AuNPs. We performed fluorescence measurements of PZPER under optimal conditions (MOPS buffer, 1 mM) at variable temperatures (Figure 3b). The spectra were recorded at 25, 50, 60, 70, 75, and 80 °C and can be used to prepare calibration curves to determine temperature based on the fluorescence of PZPER solution in MOPS buffer. Based on the obtained results, PZPER is a promising fluorescence probe for the determination of temperature. The reversibility of the developed conditions for the determination of temperature is of great interest. The spectra of PZPER solution were recorded at 25 °C, then after heating to 80 °C, and again at 25 °C after cooling. We performed three cycles, with three measurements at 25 and 80 °C (Figure 4).

The cyclic changes are reliable and reversible. The results also demonstrate that PZPER is stable at higher temperatures under aqueous conditions, which is important for the development of AuNPs soluble in water or buffer.

### 3.2. Functionalization of AuNPs with PZPER

Gold nanoparticles (AuNPs, 12 nm) were stabilized with citric acid and functionalized with PEG 3000 thiol terminated with carboxylic groups. We decided to use this thiol to stabilize AuNPs and provide a relatively large distance between the gold surface and immobilized PZPER. In this way, we attempted to avoid the potential quenching of fluorescence by gold and obtain a more efficient excitation of the chromophore. Moreover, PEG units can improve the solubility of functionalized AuNPs in aqueous solution. The terminal carboxylic groups were activated with EDCI/*N*-hydroxysulfosuccinimide sodium salt and treated with PZPER. AuNPs were purified by an Amicon filter 50 kDa to remove side products and free PZPER. The synthesis of the functionalized AuNPs is presented in Figure 5.

After purification, we collected UV–Vis and fluorescence spectra of the functionalized AuNPs in water (Figure 6). Moreover, we performed filtration of functionalized AuNPs on Amicon filters, and the filtrate was taken for measurement of fluorescence. The filtrate did not show any fluorescence, demonstrating that the AuNP solution did not contain free PZPER. The free PZPER in solution is approximately 100 times more fluorescent than PZPR connected with AuNPs (Figure 6).

The spectra of PZPER were added for comparison (Figure 6 black line). For the UV–Vis spectrum, PZPER cannot be detected for functionalized AuNPs. The main problem is the strong SPR signal from gold nanoparticles (520 nm) and the low concentration of the attached PZPER. However, the fluorescence spectrum showed a weak signal at 590 nm, which can be detected due to the fluorescence lacking gold. The weak signal resulted from the quenching of fluorescence or the insufficient excitation of the chromophore limited by gold nanoparticles (Figure 6b).

The hydrodynamic size and zeta potential of functionalized AuNPs were determined to be 23 ± 3 nm and −29 ± 4 mV, respectively, by dynamic light scattering (DLS) (Figure 7a). The negative zeta potential indicates that only a minor number of carboxylic groups were consumed for the attachment of PZPER, and most of the groups were present in an anionic form.

The TEM images showed a uniform metallic core size of 12 nm with a calculated dispersity of ±1 nm (Figure 7b). The PEG shell cannot be detected by this technique.

We also recorded fluorescence spectra for AuNPs functionalized with PZPR in MOPS buffer (1 mM, pH 5.35) at variable temperatures (Figure 8a).

The intensity of the fluorescence at 590 nm can be used to determine the temperature based on the linear calibration curve (Figure 8b). We were able to determine the amount of PZPER attached to AuNPs. The solution of AuNPs was treated at pH 1 with a HCl solution for 30 min. Then, the precipitated gold was separated, and the solution was taken for fluorescence measurement. Based on the calibration curve at pH 1, the concentration of PZPER released from AuNPs was 5 mM, which corresponds to 16% of all carboxylic groups functionalized with PZPER (see Appendix A, Appendix A).

We also verified the reversibility of the developed fluorescence probes. The fluorescence of AuNPs functionalized with PZPER was measured in MOPS buffer in three cycles between 25 and 80 °C (Figure 9).

The behavior of AuNPs functionalized with PZPER in cyclic changes in temperature is unusual. When the temperature is increased to 50 °C and then cooled to 25 °C, the fluorescence does not return to the starting intensity (black). However, in the next cycle (Figure 9, cycle 2 and 3), the signals in the experimental error are the same. The intensity of fluorescence at 50 °C is almost identical for all three cycles. Therefore, PZPER is not released when the temperature changes. When the functionalized AuNPs were purified, the fluorescence of the removed unbound PZPER was much higher than that of the remaining AuNPs with PZPER. The unusual starting fluorescence at 25 °C may occur because the PZPER folds into and disappears inside the PEG layer. When the temperature was increased and then decreased, the PZPER chromophore did not return to the starting point at which its fluorescence was partially quenched. In the next cycles, the behavior of PZPER occurred as expected. This phenomenon is reproducible with fresh AuNPs. When AuNPs were heated and cooled to room temperature, all the cycles were the same as for cycles 2 and 3.

### 3.3. Functionalization of AuNRs with PZPER

The I-Gene project (https://i-geneproject.eu/, accessed on 8 January 2024) is focused on gene editing triggered by laser light irradiation of nanotransducers. The plasmonic gold nanoparticles can absorb the light and rapidly convert it into heat via a series of photophysical processes. The gold nanorods (AuNRs) with appropriate size can be quite useful in potential in vivo applications, where tissue absorption in the near-infrared window (650–900 nm) is minimal and provides optimal light penetration. We intended to develop AuNRs functionalized with PZPER to provide a thermal probe for the quick assessment of the conversion of laser light into heat. These measurements could provide promising conditions for AuNRs based transducers developed for thermal promoted DNA cleavage. AuNRs were functionalized in similar way as AuNPs (experimental procedures are included in the Appendix A, Appendix A). The UV–Vis spectrum and TEM image are presented in Figure 10.

We performed fluorescence measurement of AuNRs functionalized with PZPER. Unfortunately, the observed fluorescence was very low. The measurement at variable temperature provided unreliable results, and the florescence did not correlate with temperature changes (Appendix A, Appendix A, Appendix A). The insufficient attachment of PZPER to AuNRs can provide low fluorescence. We determined the amount of the PZPER attached to AuNRs. The solution of AuNRs functionalized with PZPER was treated at pH 1 with HCl solution for 30 min. The precipitated gold was separated, and the solution was taken for fluorescence measurement. Based on the calibration curve at pH 1, the concentration of PZPER released from AuNPs was 0.85 μM, which corresponds to 8% of all carboxylic groups being functionalized with PZPER (see the Appendix A, Appendix A). The presence of the attached PZPER was confirmed; so, we expected the fluorescence to be much higher. We suspected that PZPER fluorescence was quenched by AuNRs. We also performed fluorescence measurement of AuNRs functionalized with thiol (HSPEG3000COOH), PZPER 0.85 μM, and a mixture of AuNRs and PZPER (Figure 11).

The AuNRs without PZPER obviously did not show any fluorescence (Figure 11 black), and the solution with PZPER (0.85 μM) demonstrated relatively high fluorescence at the concentration determined for functionalized AuNRs (Figure 11 red). However the fluorescence of the AuNRs (16.9 × 10^14^ NR/mL) and PZPER (0.85 μM) mixture was very low. It looks like the presence of gold nanorods is able to efficiently decrease the fluorescence of PZPER whether it is attached or not attached to the surface of AuNRs.

The fluorescence of PZPER is much lower (ca. 100 times) when it is attached to AuNPs or AuNRs. The concentration of attached PZPER was 5 μM for AuNPs and 0.85 μM for AuNRs. The UV–Vis spectrum of functionalized AuNRs (Figure 10a) showed plasmon peaks at 524 and 807 nm. Fluorescence spectra of AuNRs (16.9 × 10^14^ NR/mL) functionalized with PZPER were recorded with λ_ex_ = 532 nm. The overlapping of excitation wavelength with AuNRs plasmon (524 nm) and the low concentration of PZPER (0.85 μM) resulted in a lower fluorescence of AuNRs in comparison with the similar functionalized AuNPs.

## 4. Conclusions

In this work, we successfully synthesized AuNPs and AuNRs functionalized with PZPER. We demonstrated the possibility of temperature determination based on the intensity of fluorescence-developed gold nanoparticles. Based on fluorescence spectra, the intensity of emission at 590 nm can be related to the temperature, pH, and ionic strength of the environment. From this point of view, the developed functionalized AuNPs should be calibrated, and the conditions (concentration of particles, pH and ionic strength) should be strictly controlled to determine the temperature.

## Figures and Tables

**Figure 1 materials-17-01097-f001:**
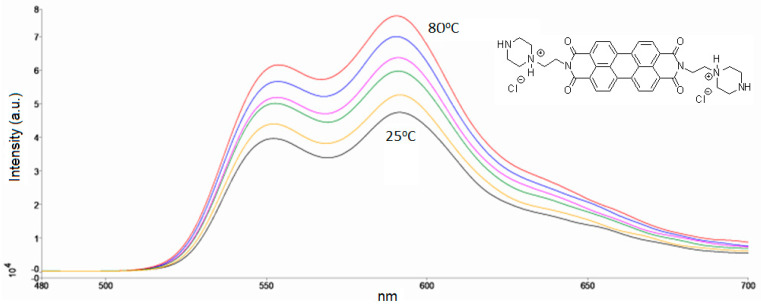
Fluorescence spectra of PZPER (50 μM, λ_ex_ = 450 nm) in water at 25, 50, 60, 70, 75, and 80 °C.

**Figure 2 materials-17-01097-f002:**
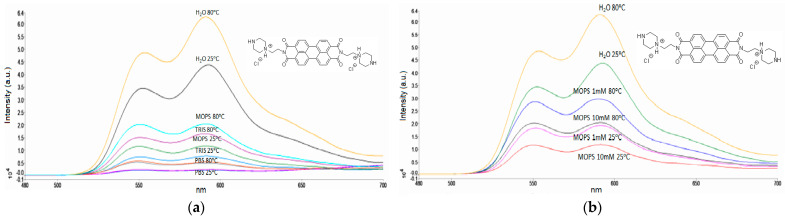
(**a**) Fluorescence spectra of PZPER (50 μM, λex = 450 nm) in water and buffers (10 mM, pH 7.4) at 25 and 80 °C, (**b**) fluorescence spectra of PZPER (50 μM, λex = 450 nm) in water and MOPS buffer (1 mM and 10 mM, pH 7.4) at 25 and 80 °C.

**Figure 3 materials-17-01097-f003:**
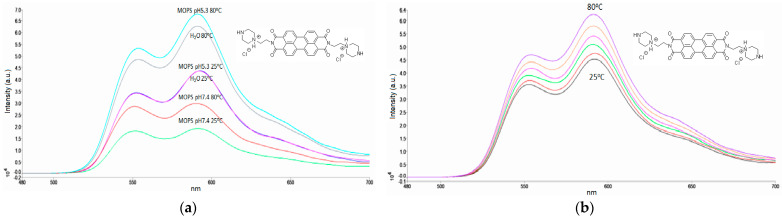
(**a**) Fluorescence spectra of PZPER (50 μM, λ_ex_ = 450 nm) in water and MOPS buffer (1 mM, pH 5.35 and 7.4) at 25 and 80 °C, (**b**) fluorescence spectra of PZPER (50 μM, λ_ex_ = 450 nm) in MOPS buffer (1 mM, pH 5.35) at variable temperatures (25 (black), 50 (red), 60 (green), 70 (purple), 75 (orange), and 80 (violet) °C.

**Figure 4 materials-17-01097-f004:**
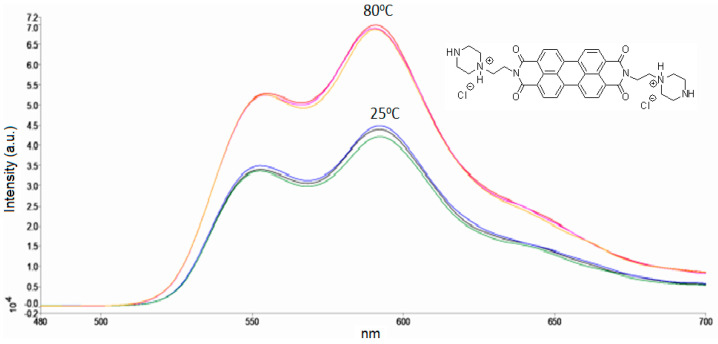
Fluorescence spectra of PZPER (50 μM, λ_ex_ = 450 nm) in MOPS buffer (1 mM, pH 5.35) at 25 (1 cycle (green), 2 cycle (blue), 3 cycle (black)) and 80 °C (1 cycle (red), 2 cycle (orange), 3 cycle (purple)) (three cycles).

**Figure 5 materials-17-01097-f005:**
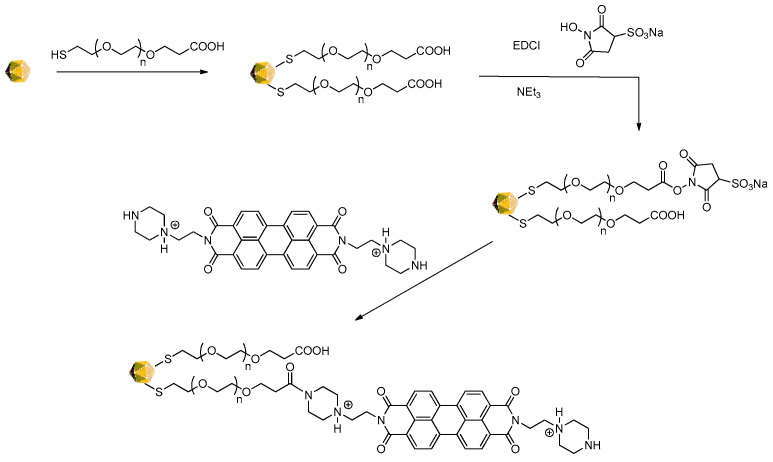
Synthesis of AuNPs functionalized with PZPER.

**Figure 6 materials-17-01097-f006:**
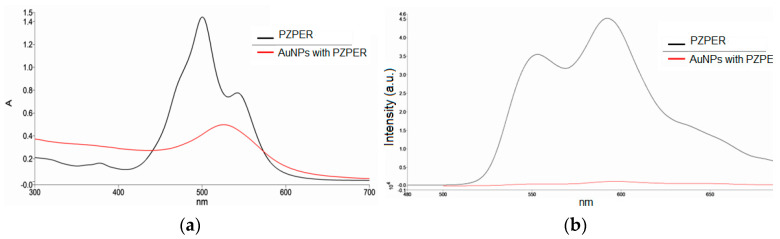
UV–Vis (**a**) and fluorescence (**b**) spectra of functionalized AuNPs (red, λ_ex_ = 450 nm) and PZPER (black, 50 μM) in water at 23 °C.

**Figure 7 materials-17-01097-f007:**
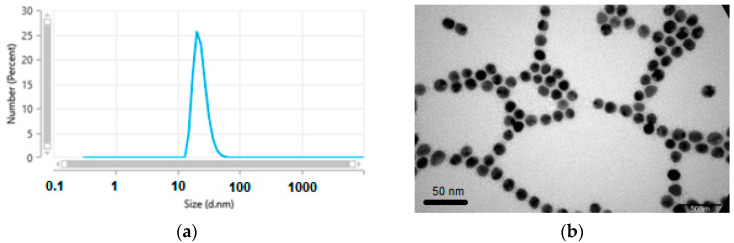
(**a**) DLS measurement of functionalized AuNPs, (**b**) TEM image of functionalized AuNPs.

**Figure 8 materials-17-01097-f008:**
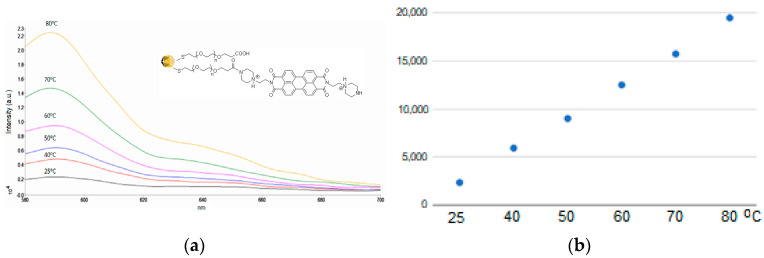
(**a**) Fluorescence spectra of AuNPs (1 × 10^12^ NP/mL) functionalized with PZPER (λ_ex_ = 532 nm) in MOPS buffer (1 mM, pH 5.35) at variable temperatures, (**b**) the correlation between the temperature and fluorescence of AuNPs functionalized with PZPER.

**Figure 9 materials-17-01097-f009:**
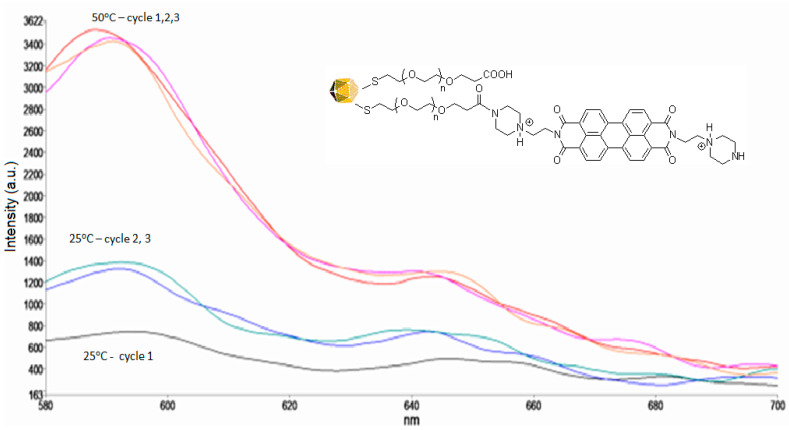
Fluorescence spectra of AuNPs (1 × 10^12^ NP/mL) functionalized with PZPER (λ_ex_ = 532 nm) in MOPS buffer (1 mM, pH 5.35) at 25 (1 cycle (black), 2 cycle (green), 3 cycle (blue)) and 50 °C (1 cycle (orange), 2 cycle (purple), 3 cycle (red)) (three cycles).

**Figure 10 materials-17-01097-f010:**
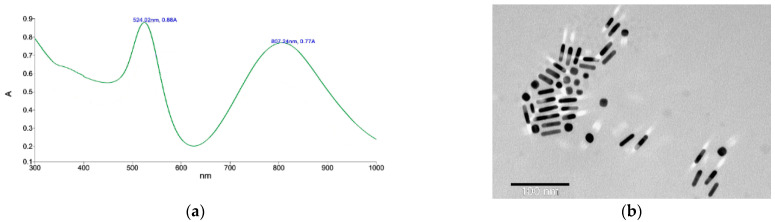
(**a**) UV–Vis spectrum of AuNRs functionalized with PZPER, (**b**) TEM image of AuNRs functionalized with PZPER.

**Figure 11 materials-17-01097-f011:**
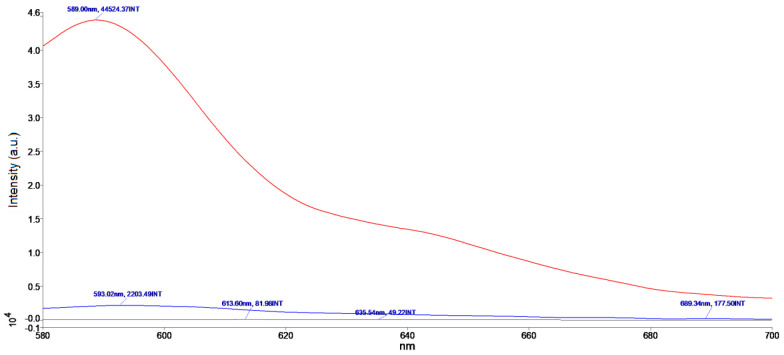
Fluorescence spectra of AuNRs (16.9 × 10^14^ NR/mL) functionalized with HSPEG3,000COOH (black), PZPER (0.85 μM) (red), and a mixture of AuNRs (16.9 × 10^14^ NR/mL) and PZPER (0.85 μM) (blue) (λ_ex_ = 532 nm) in MOPS buffer (1 mM, pH 5.35) at 25 °C.

## Data Availability

Data are contained within the article.

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
