# Peer review of "Effects Induced by the Temperature and Chemical Environment on the Fluorescence of Water-Soluble Gold Nanoparticles Functionalized with a Perylene-Derivative Dye"

_materials, 2024, doi:10.3390/ma17051097_

Round 1

Reviewer 1 Report

Comments and Suggestions for Authors

The submitted manuscript forwards the use of Au nanoparticles functionalized with a fluorescent perylene dye as a temperature dependent probe. After reviewing the manuscript, I am of the opinion that it is in need of major revisions before it could be considered for publication. Specific comments are listed below:

 (1) Lines 24-25: It is somewhat of an overstatement to say that measuring temperature can characterize “any” physical or chemical process.

 (2) The manuscript needs to be better motivated from the standpoint of the utility of a NP with a functionalized temperature dependent probe. Knowing that the Au nanoparticle itself is heated when irradiated with light makes one wonder whether the nanoparticle is altering the environment being measured.

 (3) The organization of the Materials and Methods section requires improvement as its often unclear where one subsection starts and the next ends. A complete set of italicized section headings would be helpful. Additionally, the materials list specifies reagents that are never mentioned again (e.g., AgNO3, AA) and the subsequent subsections have materials that are not listed. Lines 107-110 provide two sentences that repeat the same point. Line 111 has the phrase “the refractive index was set” but does not say to what it was set too. Terms like “shaken for 60 min” and “suitable cuvette” are vague.  At several points, the reader is referred to previously described methods. A one sentence follow-up statement that briefly describes what was done would provide improved clarity.

 (4) From the standpoint of presentation, the Result section does not meet the standards I would consider suitable for publication.  Axes on figures are often unlabelled or just provide a unit. The graph legend on Figure 6 is confusing (what is Au12) and the axes labels on Figure 6b are unreadable. Figure 11 has similar issues. The paragraphing is poor as connected thoughts are often in separate paragraphs. One sentence paragraphs usually do not appear in scholarly reports. At five points in the Results section, the reader is referred Supporting Information but where no understanding is provided as to which one of the Supporting Information Figures is being referred too.

 (5) The figure Caption to the first figure in supporting information is unclear. I do not understand what this figure is showing. Is this just the PZPER in water? The caption also refers to functionalized AuNPs where “chromophore is insufficient to observe a reliable response vs. temperature (Fig.1)” – Is this even part of the figure?

 (6) Why does the fluorescence data spectra plotted in Figure 8 and 9 start at 580 nm when other plots show the data starting at 480 nm. For both Figure 8 and 9, there is relevant data below 580 nm. Some clarification is required.

 (7) The work is motivated by biological applications but for functionalized Au nanoparticles there is little to no signal except when the pH is lowered to 5.35. This seems problematic but is never discussed.

 (8) The cycling data for 25 oC in Figure 9, as stated, shows inconsistencies between cycle 1 and cycles 2 and 3. Has this data been repeated numerous times to see if this is a reproducible phenomenon. If so, this should be stated. If not, this should be done.

 (9) When referring to the data in Figure 11, the authors stateIt looks like the presence of gold nanorods is able to efficiently quench the fluorescence of PZPER when it is attached or not to the surface of AuNRs.” How is this possible? Are you implying that fluorescence can be quenched at long distances by AuNRs?

 (10) The last sentence of the conclusion seems disconnected from the main contents of the paper.

Comments on the Quality of English Language

Needs some improvement, but for the most part is fine.

Author Response

(1) Lines 24-25: It is somewhat of an overstatement to say that measuring temperature can characterize “any” physical or chemical process.

Re: The sentence is modified. The temperature is important factor, however it does not explained everything.

“By measuring the temperature, we can at least partially characterize some physical or chemical process in the living or still environment.”

(2) The manuscript needs to be better motivated from the standpoint of the utility of a NP with a functionalized temperature dependent probe. Knowing that the Au nanoparticle itself is heated when irradiated with light makes one wonder whether the nanoparticle is altering the environment being measured.

Re: The plasmonic gold nanoparticles can absorb the light and rapidly convert it into heat via a series of photophysical processes. However, the wavelength of irradiated light should resonate with plasmon of gold nanoparticles. The developed AuNPs and AuNRs can be used to determined temperature when sample is irradiated with appropriate wavelength or without irradiation. In the first option the temperature measurement reflects the produced heat from interaction of gold nanoparticles with light. In the second option the observed fluorescence corresponds to temperature of solution. Our motivation is expressed in the lines 262-270.

(3) The organization of the Materials and Methods section requires improvement as its often unclear where one subsection starts and the next ends. A complete set of italicized section headings would be helpful. Additionally, the materials list specifies reagents that are never mentioned again (e.g., AgNO3, AA) and the subsequent subsections have materials that are not listed. Lines 107-110 provide two sentences that repeat the same point. Line 111 has the phrase “the refractive index was set” but does not say to what it was set too. Terms like “shaken for 60 min” and “suitable cuvette” are vague.  At several points, the reader is referred to previously described methods. A one sentence follow-up statement that briefly describes what was done would provide improved clarity.

Re: The italicized section are added. The list of reagents comprises compounds used for preparation of AuNPs and AuNRs. The procedure provided in supporting information required some of these reagents.

Line 107-110 were converted to one sentence. The refractive index was added.

Procedures are clarified and additional sentences for described methods are added.

(4) From the standpoint of presentation, the Result section does not meet the standards I would consider suitable for publication.  Axes on figures are often unlabelled or just provide a unit. The graph legend on Figure 6 is confusing (what is Au12) and the axes labels on Figure 6b are unreadable. Figure 11 has similar issues. The paragraphing is poor as connected thoughts are often in separate paragraphs. One sentence paragraphs usually do not appear in scholarly reports. At five points in the Results section, the reader is referred Supporting Information but where no understanding is provided as to which one of the Supporting Information Figures is being referred too.

Re: The figures legends and labeling is corrected. The details about supporting information referred to the text is clarified.

(5) The figure Caption to the first figure in supporting information is unclear. I do not understand what this figure is showing. Is this just the PZPER in water? The caption also refers to functionalized AuNPs where “chromophore is insufficient to observe a reliable response vs. temperature (Fig.1)” – Is this even part of the figure?

Re: The supporting information is modified to avoid confusion. Figure S1 is just UV-Vis spectra of the PZPER in water at variable temperature.

(6) Why does the fluorescence data spectra plotted in Figure 8 and 9 start at 580 nm when other plots show the data starting at 480 nm. For both Figure 8 and 9, there is relevant data below 580 nm. Some clarification is required.

Re: The fluorescence spectra of PZPER Figure 1-4 are recorded for excitation λex = 450 nm. However Figure 8 and 9 presented fluorescence spectra of AuNPs functionalized with PZPER were recorded with excitation λex = 532 nm.

(7) The work is motivated by biological applications but for functionalized Au nanoparticles there is little to no signal except when the pH is lowered to 5.35. This seems problematic but is never discussed.

Re: The response of thermal probe based on the AuNPs functionalized with PZPER is more complex. The same pH (7.4) for different buffers (MOPS, TRIS, PBS) provided different intensity of fluorescence (Figure 2a). The concentration of MOPS buffer also affected the fluorescence (Figure 2b). From this point of view the most problematic part of measurement is the calibration of thermal probe based on the fluorescence intensity changes.

(8) The cycling data for 25 oC in Figure 9, as stated, shows inconsistencies between cycle 1 and cycles 2 and 3. Has this data been repeated numerous times to see if this is a reproducible phenomenon. If so, this should be stated. If not, this should be done.

Re: This experiment was repeated several times. This phenomenon is reproducible with fresh AuNPs. When AuNPs were heated and cool to room temperature then all cycles were the same as for cycles 2 and 3.

(9) When referring to the data in Figure 11, the authors state “It looks like the presence of gold nanorods is able to efficiently quench the fluorescence of PZPER when it is attached or not to the surface of AuNRs.” How is this possible? Are you implying that fluorescence can be quenched at long distances by AuNRs?

Re: We observed that fluorescence of PZPER is decreased when it is attached to the AuNRs or AuNRs (without PZPER) are added to the PZPER solution (0.85 microM). We are not implying that fluorescence can be quenched at long distances by AuNRs. The observed low fluorescence for both cases resulted from relatively low PZPER concentration (0.85 microM) and overlapping of excitation wavelength (λex = 532 nm) with AuNRs plasmon (524 nm).

(10) The last sentence of the conclusion seems disconnected from the main contents of the paper.

Re: The sentence is removed.

Reviewer 2 Report

Comments and Suggestions for Authors

Agnieszka Lindstaedt et al reported a manuscript entitled ‘Effects induced by the temperature and chemical environment 2 on the fluorescence of water soluble gold nanoparticles func-3 tionalized with a perylene derivative dye’ where they mentioned a development a fluorescent molecular probe based on gold nanoparticles functionalized 12 with N,N’-bis(2-(1-piperazino)ethyl)-3,4,9,10-perylenetetracarboxylic acid diimide dichloride for potential applications. The paper is interesting. Here are my comments on the following:

v  Figure 1 has to be modified. You can put wavelength as an X-axis and make it larger so others can see properly. Y axis should be written as Intensity (a.u.). Also, the font size will be large.

v  Why is the intensity increased with temperature in Fig 1?

v  The authors should include the absorption spectra and change in Fig 1 with the PL intensity figure.

v  What is the cause of both emission at 540 nm and ~590 nm?

v  There is no label (Wavelength (nm) in the Figure 2a. Please check it

v  Do the perfect labelling for Fig 1 to Fig. 4.

v  Fig. 6 needs to be improved. Also, mention a and b labels.

v  Authors can mention more clearly why Au NR did not show any PL emission where it was different for the case of Au NP.

v  Due to the fluorescence intensity changes exhibited by modified gold nanoparticles. Optical properties could be changed with shape and size variation and with doping. Authors could cite a few references in the introduction like (i) 10.1515/nanoph-2022-0797 (ii) 10.1515/nanoph-2022-0503

Comments on the Quality of English Language

Please check it carefully to avoid any mistakes. 

Author Response

We have considered all reviewer comments and a point-by-point response is provided:

(1) Figure 1 has to be modified. You can put wavelength as an X-axis and make it larger so others can see properly. Y axis should be written as Intensity (a.u.). Also, the font size will be large.

Re: The recommended modifications have been applied.

(2) Why is the intensity increased with temperature in Fig 1?

Re: The increased fluorescence intensity of PZPER with temperature was already described. According to literature, this behavior suggests the formation of H-aggregates of dyes due to the p–p stacking of chromophores in a ladder fashion, also at very low dye concentration. The ratio between monomer and H-type aggregates is temperature dependent and as the results the temperature dependent fluorescence spectra are observed. (F. Donati, A. Pucci, C. Cappelli, B. Mennucci and G. Ruggeri, J. Phys. Chem. B, 2008, 112, 3668–3679; F. Donati, A. Pucci, G. Ruggeri, Phys. Chem. Chem. Phys. 2009, 11, 6276–6282).

(3) The authors should include the absorption spectra and change in Fig 1 with the PL intensity figure.

Re: The UV-Vis spectra of PZPER at variable temperature are included in supporting information Figure S1. However, the UV-Vis spectra are insufficient for gold nanoparticles functionalized with PZPER. The low concentration of PZPER at the surface and strong plasmon of gold did not allowed to corelate temperature changes with UV-Vis spectra of AuNPs functionalized with PZPER.

(4) What is the cause of both emission at 540 nm and ~590 nm?

Perylene diimide dyes are characterized by the same absorption and emission features due to the presence of nodes in the HOMO and LUMO at the imide nitrogen which avoid the electronic coupling between the perylene nuclei and the imide substituents. The fluorescence spectra for PZPER aqueous solutions displayed emission maxima pointed at about 540 and 590 nm attributed to 0–0 and 0–1 relaxations of vibronic transitions, respectively. It was already described: lit. 52 Donati, F.; Pucci, A.; Ruggeri, G. Temperature and chemical environment effects on the aggregation extent of water soluble perylenedye into vinyl alcohol-containing polymers. Phys. Chem. Chem. Phys. 2009, 11, 6276–6282.

(5) There is no label (Wavelength (nm) in the Figure 2a. Please check it

Re: The appropriate label is added.

(6) Do the perfect labelling for Fig 1 to Fig. 4.

Re: The appropriate labels are added.

(7) Fig. 6 needs to be improved. Also, mention a and b labels.

Re: The appropriate labels are added.

(8) Authors can mention more clearly why Au NR did not show any PL emission where it was different for the case of Au NP.

Re: We added clear comment why AuNRs demonstrated much lower emission than AuNPs. (line 299-305).

(9) Due to the fluorescence intensity changes exhibited by modified gold nanoparticles. Optical properties could be changed with shape and size variation and with doping. Authors could cite a few references in the introduction like (i) 10.1515/nanoph-2022-0797 (ii) 10.1515/nanoph-2022-0503

Re: The suggested references are dealing with:

(i) Two-dimensional metal halide perovskites and their heterostructures: from synthesis to applications 10.1515/nanoph-2022-0797

(ii) Exploring magneto-optic properties of colloidal two-dimensional copper-doped CdSe nanoplatelets 10.1515/nanoph-2022-0503

The introduction is devoted to  recent developments of thermal probes based on the fluorescence. From this point of view the suggested references do not comprise the scope of introduction.

Round 2

Reviewer 1 Report

Comments and Suggestions for Authors

The authors have adequately address my concerns.

Comments on the Quality of English Language

Moderate editing required